# The production of national defense and the macroeconomy

José L. Torres  *

Department of Economics, University of Malaga, Malaga, Spain

* jtorres@uma.es

## Abstract

This study investigates the interactions between defense production and the rest of the economy. We develop a two-sector dynamic stochastic general equilibrium model with military and nonmilitary production. Inputs (capital and labor) are distributed between the two sectors. Calibration of the model to key targets of the US economy results in an elasticity of substitution between consumption of goods and services and national defense services of 0.56. The estimated complementarity between consumption goods and defense services results in positive spillovers across military and final goods production sectors, even when the nonmilitary production function is not directly related to military spending. We find that military spending is procyclical and that military spending as a percentage of output is countercyclical. Finally, investment-specific technological shocks to military equipment have a positive impact on nonmilitary output, although they reduce business investment.

**Data Availability Statement:** All relevant data are within the manuscript.

**Funding:** I acknowledge financial support from the University of Malaga and project financed by the Spanish Ministry of Science and Technology ECO2016-76818-C3-2-P. The funders had no role in study design, data collection and analysis,

## Introduction

National defense is a valuable public good, and consequently a fraction of total resources of the economy are expended on military activities for producing defense services. All countries, with some particular exceptions, devote a fraction of income for military spending, with the aim of providing a sufficient level of national security. According to the Stockholm International Peace Research Institute (SIPRI), military spending as a percentage of GDP in 2018 ranged from 0% in some small countries to 20.7% in Eritrea. Military personnel reached a maximum in North Korea, according to the International Institute for Strategic Studies (IISS), with a military force of 50.5 per 1,000 inhabitants. As Hartley [1] pointed out, military spending has the objective of improving a country's ability in defending its national interests against potential threats, and therefore, a demand for national defense exists. A major policy determination for governments is the issue of how to allocate resources to maximize social welfare, and hence, the decision on the optimal fraction of resources for military spending. However, there is no consensus in the literature about the relationship between economic activity and military expenditure, and most recent evidence suggests no relationship between military expenditure and the economy [2].

A variety of models have been developed in the literature to explain the demand for military expenditure and the production of defense, based on the neoclassical model by the inclusion

decision to publish, or preparation of the
manuscript.

**Competing interests:** The authors have declared
that no competing interests exist.

of national defense in a market economy with public goods. The contributions differ in how
national defense is produced and on the effects of military spending on total output, but all of
them have the common characteristic that security is an argument in the social welfare func-
tion. In these kinds of models the arguments of the social welfare functions are assumed to be
civilian output and military capital stock or national security. National defense is a function of
military spending, aggregate output, and other strategic variables representing the potential
external threat. These theoretical developments have been put forward to explain the determi-
nation of military expenditures. Examples are [3–8]. An alternative approach, based on the
arms race literature, is to consider that the domestic weapons stock is an argument of the util-
ity function [9–14].

Another branch of the literature has focused on the relationship between military spending
and economic growth. There is a large number of papers studying the effects of military spend-
ing on economic growth. Nevertheless, the existing empirical literature yields ambiguous
results about the effects of military spending on economic growth, not only in the size but also
on the sign. This literature started with the works of Benoit [15, 16], which found a positive
relationship between defense spending and economic growth for 44 developing countries for
the period 1950–1965. The empirical literature investigating the relationship between military
spending and economic growth has grown exponentially, for a number of countries and peri-
ods, leading in general to the estimation of a positive link between growth and military spend-
ing. This empirical literature has used several alternative specifications of the aggregate
production function to study the effects of military expenditure on economic growth, but
most of them have been based on the Feder-Ram model [2, 17]. Dunne et al. [17] reviewed this
empirical literature and showed that while the growth literature has not found military expen-
diture to be a significant determinant of economic growth, the defense economics literature
has found positive effects, with the difference explained by the different theoretical model used
for the reduced estimation.

This study contributes to the literature by investigating the interactions between military
spending and the rest of the economy across the business cycle in a two-sector neoclassical-
economy framework. Here we follow the first approach and consider national defense as an
argument in the household's utility function. The paper develops a defense-dynamic stochastic
general equilibrium (D-DSGE) model with two production sectors: one produces consump-
tion and investment goods, and the other produces national defense. Both technologies use a
combination of capital and labor inputs, and a stand-in agent's utility function includes
national security as a public good. That is, we adopt the view that defense spending provides
protection to a nation's citizens, who equally benefit from it. In this context, we specify a pro-
duction function for defense where it is assumed that a benevolent government chooses mili-
tary spending to maximize the welfare of the representative private agent. Defense technology
uses both labor (military personnel) and capital (military structures and equipment) inputs to
produce defense services. This specification implies that a fraction of total investment is allo-
cated to military capital assets investment, and also a fraction of total labor is allocated to
working in defense production activities. The model includes four technological shocks: a neu-
tral technological shock to each production technology, and an investment-specific technolog-
ical change to each type of physical capital.

First, we calibrate the model by matching some observed features for the US economy in
order to estimate the value of the parameters, both preference and technological, related to
the defense sector. A key issue in assessing the relationship between the military sector and
the rest of the economy is the complementary/substitutability between consumption goods

and services and defense. In the literature, public goods, such as public education and public health services are considered as a complement of private goods, and some degree of substitution is considered between both types of goods. However, other public goods are much more difficult to be substitutes of private goods, as would be the case for national defense. Barro [18] argued that national defense provides very little substitution with private consumption. By contrast, other authors have treated both goods as substitutes. Smith [3] concluded that the elasticity of substitution between security and civilian output is quite high. For the UK, [3] estimated a value of 4.87, indicating that security decisions are very sensitive to the cost of provision. Evans and Karras [19] found that the degree of substitutability between government services and private consumption depends negatively on the fraction of government spending on national defense, suggesting that national defense and private consumption are highly complementary. In this study, we find that the elasticity of substitution between private consumption of goods and services and defense services is very low, indicating that they are complementary. Calibration of the model for the US economy results in an elasticity of 0.56, well below one.

Second, we pay attention to the relationship between the military sector and the rest of the economy under two different technological shocks: neutral shocks specific to each sector, and investment-specific technological change (ISTC) shocks specific to each capital asset. We find that the relationship between the military sector and the rest of the economy is positive over the business cycle. Military spending and the production of defense is procyclical, but military spending responds less than proportionally with respect to output, and hence, military spending as a percentage of GDP is found to be countercyclical. The most striking result is that the model predicts positive spillovers across both sectors. This result is robust, as our model does not include any ad hoc specification for output where military spending enters as an additional input, as is common in the economic defense literature. However, a positive aggregate productivity shock affecting the military sector has a small impact on the rest of the economy, given that the military sector is a very small fraction of the total economy. This shock can be interpreted as a decline in the potential threat, which increases the production of national defense given a certain amount of military spending in military equipment and personnel. Next, we identify the relationship between both sectors over the business cycle when specific technological shocks have an impact on the technology embodied in new vintage capital equipment incorporated via the investment process in both sectors. When ISTC occurs in the business capital assets, we find a positive impact on economic activity, as expected, but also in the military sector, as military investment increases. Similarly, when an ISTC shock affects military equipment, we find that the final effect on the economy is positive, although this shock has a negative effect on business investment. In both cases, ISTC shocks have a negative impact on consumption of goods and services. Therefore, fostering investment in R&D activities in the military sector diverts resources from civilian investment to military investment, reducing the nonmilitary capital accumulation process, although the final impact on the economy is positive.

The rest of the paper is organized as follows. The material and methods section presents a two-sector DSGE model where the defense production function is defined, and the external calibration of the parameters of the model. The results and discussion section presents the structural estimated values for preference and technological parameters of the military sector, the results of the simulations of neutral technological shocks to each sector, and the results of the effects of ISTC to both nonmilitary and military capital assets. Finally, the last section presents our conclusions.

## Materials and methods

### A defense-dynamic stochastic general equilibrium model

The paper considers a two-sector defense-dynamic stochastic general equilibrium (D-DSGE) model that includes national security (the production of defense services) in the utility function of a representative household. Defense is considered a public good, producing utility additional to private goods and services, as well as leisure. National security services is an output which is produced by operating a technology that takes military spending in structures and equipment as well as in labor (military personnel) as inputs. In this framework, social welfare depends not only on private consumption and leisure but also on national security services. This theoretical framework allows us to determine the optimal military spending that maximizes social welfare.

We find the optimal allocation of consumption and national security solving the central planner problem, whose objective is to maximize social welfare. Alternatively, we can assume a market environment, where agents make decisions on consumption and investment and the government decides the level of military expenditures. If these expenditures are financed with non-distortionary taxes, both situations are equivalent (taking into account the agents' preferences, and given that the objective function of the central planner is to maximize social welfare). The central planner chooses the optimal level of military expenditure in order to produce the optimal level of national security considering the agent's preferences. The role of the central planner is to balance the welfare benefits of extra security derived from military expenditure with its opportunity cost in foregone consumption and investment.

We assume that total output (GDP), $Y_t$, is defined by the following feasibility constraint of the economy,

$$Y_t = C_t + I_{k,t} + M_t \tag{1}$$

indicating that total output of the economy can be used in three ways: consumption of goods and services in the nonmilitary sector, $C_t$, investment in fixed assets in the nonmilitary (business) sector, $I_{k,t}$, and military spending, $M_t$, as specified by [2]. Importantly, this definition of output implies that the military spending component only includes investment in military physical capital and military consumption of goods and services, excluding compensation to military personnel. The military spending component in the definition of DGP using the expenditure approach does not include compensation to military personnel to avoid double accounting. Total military spending, denoted by $M_t^*$, is devoted to investment in military structures and equipment, $I_{m,t}$, military consumption of goods and services, $C_{m,t}$, and for paying salaries, $W_{m,t}$, to military personnel denoted by $S_t$, such as

$$M_t^* = C_{m,t} + I_{m,t} + W_{m,t}S_t \tag{2}$$

From national accounts basic identities, GDP using the income approach, consistent with our model, is defined as total compensation to employees in both the final production sector and the military sector, plus gross capital income

$$Y_t = W_{k,t}L_t + W_{m,t}S_t + R_tK_t \tag{3}$$

where $W_{k,t}L_t$ is the compensation to employees in the final goods sector, and $R_tK_t$ is capital income in the nonmilitary production sector, as there are no capital gains from the military capital stock. By defining $M_t = M_t^* - W_{m,t}S_t$, it results that:

$$Y_t = C_t + I_{k,t} + C_{m,t} + I_{m,t} \tag{4}$$

where $M_t = C_{m,t} + I_{m,t}$, given that both definitions of output (expenditure approach and income approach) must be equal.

**Households.** We assume that households' utility function is a function of consumption of goods and services, $C_t$, defense services, $D_t$, working hours in the final goods sector, $L_t$, and working hours in the military sector, $S_t$:

$$U_t = U(C_t, D_t, L_t, S_t) \tag{5}$$

where the instantaneous utility function $U(\cdot)$ is a nonnegative function, concave and twice differentiable. In our model, we consider national defense services as an argument of the utility function, and the labor force is split between the final goods productive sector and the defense sector (military personnel). Total available discretionary time is normalized to one, and leisure, denoted by $O_t$, is defined as:

$$O_t = 1 - L_t - S_t \tag{6}$$

In particular, we consider that consumers' preferences are given by the following instantaneous utility function:

$$U(C_t, D_t, L_t, S_t) = \gamma ln(\omega C_t^\eta + (1 - \omega)D_t^\eta)^{1/\eta} + (1 - \gamma)\ln(1 - L_t - S_t) \tag{7}$$

where total consumption is defined as a composite CES function of private consumption of goods and services and national defense services. The parameter $\omega (0 < \omega < 1)$ measures the weight of consumption of goods and services relative to defense in the utility function, and $\eta$ $(-\infty, 1)$ indicates the elasticity of technical substitution between consumption of goods and services and defense. We can define $\sigma = 1/(1 - \eta)$ as the measure of the intratemporal elasticity of substitution between nonmilitary consumption and defense. When $\sigma = 0 (\eta = -\infty)$, both types of goods are complementary, whereas when $\eta = 1 (\sigma = \infty)$ both types of goods are perfect substitutes.

The budget constraint faced by the representative household is:

$$C_t + I_{k,t} + C_{m,t} + I_{m,t} = Y_t \tag{8}$$

where $I_{k,t}$ is business capital investment, $C_{m,t}$ is military consumption of good and services, and $I_{m,t}$ is military investment in military equipment and structures. This budget constraint implies that total output can be distributed among consumption and investment in the two sectors. We assume two independent capital accumulation processes, as few physical capital transfers between the two sectors can be produced. Military and nonmilitary equipment have different characteristics that make asset reallocation difficult between both sectors (it is hard to think of a civil use for fighter aircraft), although structures can be more easily adapted (for instance the conversion of an air base facility to a civil airport).

Installing new capital involves adjustment costs. These investment adjustments are assumed to be quadratic [20], for $i = k, m$:

$$\frac{\varphi_i}{2}\left(\frac{I_{i,t}}{I_{i,t-1}} - 1\right)^2 \tag{9}$$

Nonmilitary (business) capital holdings, $K_t$, change over time according to:

$$K_{t+1} = (1 - \delta_k)K_t + Z_{k,t}I_{k,t}\left[1 - \frac{\varphi_k}{2}\left(\frac{I_{k,t}}{I_{k,t-1}} - 1\right)^2\right] \tag{10}$$

where $\delta_k$ is the business physical capital depreciation rate, and $Z_{k,t}$ is an ISTC shock representing technological progress embodied in the new capital assets that are incorporated into the economy. Similarly, the accumulation of military physical capital, $X_t$, is defined as:

$$X_{t+1} = (1 - \delta_m)X_t + Z_{m,t}I_{m,t}\left[1 - \frac{\varphi_m}{2}\left(\frac{I_{m,t}}{I_{m,t-1}} - 1\right)^2\right] \tag{11}$$

where $\delta_m$ is the military physical capital depreciation rate and $Z_{m,t}$ is the ISTC to military physical capital representing technological progress embodied in new military physical assets. ISTC can be different between military and nonmilitary capital assets. We assume that both types of ICTC follow exogenous stochastic processes given by:

$$lnZ_{i,t} = (1 - \rho_i)Z_i + \rho_i lnZ_{i,t-1} + \varepsilon_t^{i,z} \; for \; i = k, m \tag{12}$$

where $Z_i$ is the steady-state value for the (inverse) relative price of investment for the two capital assets, $\rho_i < 1$ is the autoregressive parameter, and $\varepsilon_t^{i,z}$ is a random i.i.d. innovation in the stochastic process.

**Aggregate production function.** A key element of the model is the election of the appropriate form of the aggregate production function and its relationship with military expenditure. Studies in the literature have included military spending in the final output production function using alternative specifications (see [6] and [10]). Biswas and Ram [21] included military spending as an additional input in the final output production function. Knight et al. [22] included military spending as a component in an otherwise standard output growth equation and found that military spending has a negative effect on capital formation and resource allocation. Gong and Zou [14] used alternative specifications by considering military spending as a consumption good or as an investment good. Aizeman and Glick [23] argued that output is influenced by national security, where national security depends on military expenditure relative to the threat. In this context, military expenditure induced by external threats should increase output while military expenditure not induced by external threats (and induced by rent seeking and/or corruption) should reduce output. Dunne et al. [17] carried out a critical review of models of military expenditure, indicating that the resulting relationship between economic growth and military spending largely depends on the particular specification of the production function, and the inclusion or not of a military spending component affecting final output directly as an additional input or indirectly by affecting some technological component of the production function. However, all specifications, including military expending in the aggregate production function, are rather ad hoc, and little justification has been provided by the literature. An alternative and arguably more realistic way to model the effect of military spending on output is to consider a binary variable multiplying the production function. With zero military spending, national defense production is also zero, and any potential threat would have dramatic negative effects on production activities. When military spending is positive and a threshold minimum value for national defense is reached, final output can be produced in a security environment. Once this threshold value is achieved, additional military expenditures have no further direct positive effects on output.

To keep the model as close as possible to the standard neoclassical one-sector model, we assume that total aggregated output of the economy, $Y_t$, is obtained from capital and labor:

$$Y_t = A_t K_t^\alpha L_t^{1-\alpha} \tag{13}$$

where $K_t$ is nonmilitary capital stock, $L_t$ is nonmilitary employment, and $A_t$ is total factor (neutral-technology) productivity in the final goods production sector. Total factor productivity

(TFP) is assumed to follow an exogenous stochastic process given by:

$$lnA_t = (1 - \rho_A)A + \rho_A lnA_{t-1} + \varepsilon_t^A \qquad (14)$$

where $A$ is the steady-state value for TFP, $\rho_A < 1$, and $\varepsilon_t^A$ is a random i.i.d. innovation in the stochastic process.

Therefore, the representative firm operates a standard Cobb-Douglas production function. One key point of this specification is that we assume that defense spending does not appear as an argument of the final output of the economy; that is, we assume the independence of the final goods and services production function from military spending. Military spending influences final output only through the general equilibrium effects on nonmilitary capital and labor. This implies that military spending is similar to other forms of government spending. Our choice of this approach is based on empirical evidence for the impact of government spending on final output: defense spending only diverts resources away from the civil sector of the economy. However, general equilibrium effects of military spending on investment in physical capital and military personnel will affect final output. Here we focus on the general equilibrium effects rather than assuming a direct effect at the outset.

**The military production function.** The military production function defines the technology for producing defense services. National defense is considered as a public nonrival nonexcludable good, and a fraction of inputs endowment is used for producing this public good. In this general equilibrium setup we need to specify the production function of national security as a function of the resources of the economy (produced by the nonmilitary sector) taken as inputs [3]. However, the optimal allocation of resources yields an expenditure function. In other words, military spending can be reformulated and written as the result of minimizing a loss function given the technical transformation of military expenditure in the provision of security, as it is assumed that policy makers would like to divert to the military sector as few resources as possible.

We assume a CES technology for the production of defense services, $D_t$, given by:

$$D_t = B_t[\mu X_t^\phi + (1 - \mu)S_t^\phi]^{1/\phi} \qquad (15)$$

where $X_t$ is the stock of military capital assets (structures and equipment), $S_t$ is the labor input (military personnel), and $B_t$ is a neutral-technical shock to the production of defense services, similar to the TFP shock of the rest of the economy. The parameter $\mu(0 < \mu < 1)$ is the proportion of military capital in the defense production function, and $\phi$ is a parameter driving the substitutability between military capital (military structures and equipment) and military personnel.

We assume that the neutral technological shock follows an exogenous stochastic process given by:

$$lnB_t = (1 - \rho_B) + \rho_A lnB_{t-1} + \varepsilon_t^B \qquad (16)$$

where $B$ is the steady-state value of aggregate productivity in the military sector, $\rho_B < 1$, and $\varepsilon_t^B$ is a random i.i.d. innovation in the stochastic process.

**Central planner.** In this theoretical framework, the households' and firms' maximization problems are equivalent to the problem faced by the central planner, which is to maximize the value of the representative consumer lifetime utility subject to the economy budget constraint and the defense production function, where initial capital stock $K_0$ and $X_0$ are given, $E_t(\cdot)$ is the conditional expectation operator evaluated at time 0, and $\beta \in (0,1)$ is the consumer's

discount factor. The maximization problem faced by the central planner is:

$$\max_{\{C_t,D_t,L_t,S_t,K_t,X_t,I_{k,t},I_{m,t}\}} E_t \sum_{t=0}^{\infty} \beta^t \left[\frac{\gamma}{\eta} \ln\left(\omega C_t^{\eta} + (1-\omega)D_t^{\eta}\right) + (1-\gamma)\ln(1-L_t-S_t)\right] \quad (17)$$

subject to the restrictions given by (8), (10), (11), (13), and (15), and where $K_0$, $X_0$ are given.

From the first-order conditions, we obtain the following conditional demand for consumption and for security services, respectively:

$$\lambda_{1,t} = \beta^t \gamma \omega (\omega C_t^{\eta} + (1-\omega)D_t^{\eta})^{-1} C_t^{\eta-1} \quad (18)$$

$$\lambda_{2,t} = \beta^t \gamma (1-\omega)(\omega C_t^{\eta} + (1-\omega)D_t^{\eta})^{-1} D_t^{\eta-1} \quad (19)$$

where $\lambda_1, t$ is the shadow price of private consumption of goods and services, and $\lambda_{2,t}$ is the shadow price of national defense services. The relative shadow price of private consumption to defense services is given by:

$$\frac{\lambda_{1,t}}{\lambda_{2,t}} = \frac{\omega C_t^{\eta-1}}{(1-\omega)D_t^{\eta-1}} \quad (20)$$

The equilibrium relative shadow price depends on the distribution parameter $\omega$ and the parameter governing the elasticity of substitution, $\eta$. On the other hand, Lagrange's multipliers $\lambda_{3,t}$ and $\lambda_{4,t}$ are the Tobin's Q marginal rate, defined as $q_{i,t}$, for each capital asset, respectively, multiplied by the shadow price of private consumption of goods and services, such as,

$$q_{k,t} = \frac{\lambda_{3,t}}{\lambda_{1,t}} \quad (21)$$

and

$$q_{m,t} = \frac{\lambda_{4,t}}{\lambda_{1,t}} \quad (22)$$

Equilibrium conditions for private consumption and the defense services optimal path are, respectively,

$$\frac{(\omega C_{t+1}^{\eta} + (1-\omega)D_{t+1}^{\eta})^{-1} C_{t+1}^{\eta-1}}{(\omega C_t^{\eta} + (1-\omega)D_t^{\eta})^{-1} C_t^{\eta-1}} \frac{\beta Z_{k,t}}{q_{k,t}} \left[\frac{q_{k,t+1}(1-\delta_k)}{z_{k,t+1}} + \alpha \frac{Y_{t+1}}{K_{t+1}}\right] = 1 \quad (23)$$

$$\frac{(\omega C_{t+1}^{\eta} + (1-\omega)D_{t+1}^{\eta})^{-1} D_{t+1}^{\eta-1}}{(\omega C_t^{\eta} + (1-\omega)D_t^{\eta})^{-1} D_t^{\eta-1}} \frac{\beta Z_{m,t}}{q_{m,t}} \left[\frac{q_{m,t+1}(1-\delta_m)}{z_{m,t+1}} + \mu B_t D_t^{1-\phi} Z_t^{\phi-1}\right] = 1 \quad (24)$$

Equilibrium conditions for nonmilitary and military investment, respectively, are given by:

$$q_{k,t}\left[1 - Z_{k,t}\frac{\varphi_k}{2}\left(\frac{I_{k,t}}{I_{k,t-1}} - 1\right)^2 - Z_{k,t}\varphi_k\left(\frac{I_{k,t}}{I_{k,t-1}} - 1\right)\frac{I_{k,t}}{I_{k,t-1}}\right] +$$

$$\beta\frac{(\omega C_t^\eta + (1-\omega)D_t^\eta)^{-1}C_t^{\eta-1}}{(\omega C_{t+1}^\eta + (1-\omega)D_{t+1}^\eta)^{-1}C_{t+1}^{\eta-1}}q_{k,t+1}\left(\frac{I_{k,t}}{I_{k,t-1}} - 1\right)\left(\frac{I_{k,t}}{I_{k,t-1}}\right)^2 = 1 \quad (25)$$

$$q_{m,t}\left[1 - Z_{m,t}\frac{\varphi_m}{2}\left(\frac{I_{m,t}}{I_{m,t-1}} - 1\right)^2 - Z_{m,t}\varphi_m\left(\frac{I_{m,t}}{I_{m,t-1}} - 1\right)\frac{I_{m,t}}{I_{m,t-1}}\right] +$$

$$\beta\frac{(\omega C_t^\eta + (1-\omega)D_t^\eta)^{-1}C_t^{\eta-1}}{(\omega C_{t+1}^\eta + (1-\omega)D_{t+1}^\eta)^{-1}C_{t+1}^{\eta-1}}q_{m,t+1}\left(\frac{I_{m,t}}{I_{m,t-1}} - 1\right)\left(\frac{I_{m,t}}{I_{m,t-1}}\right)^2 = 1 \quad (26)$$

Finally, equilibrium labor conditions for workers and military personnel are given by:

$$\frac{(1-\gamma)}{(1 - L_t - S_t)} = \frac{\gamma\omega C_t^{\eta-1}(1-\alpha)Y_t}{(\omega C_t^\eta + (1-\omega)D_t^\eta)L_t} \quad (27)$$

$$\frac{(1-\gamma)}{(1 - L_t - S_t)} = \frac{\gamma(1-\omega)D_t^\eta B_t(1-\mu)S_t^{\phi-1}}{(\omega C_t^\eta + (1-\omega)D_t^\eta)D_t^\phi} \quad (28)$$

## Calibration

The model includes preference and technological parameters from both sectors. For the nonmilitary sector parameters enough information is available, and we choose standard values used in the literature. However, for the military sector no a priori information is available regarding the values of these parameters. This is critical, as the model includes one key parameter driving the relationship between both sectors: the elasticity of substitution between consumption goods and services and the public good comprising national security services. Therefore, in the calibration of the model we use the following strategy. For nonmilitary sector parameters, values are taken from the literature. For military sector parameters, they are internally calibrated using the steady-state equations of the model to match some key targets. The model is calibrated for the US economy.

First, we calibrate externally the set of parameters ($\beta$, $\alpha$, $\gamma$, $\omega$, $\delta_k$, $\delta_m$). For the discount factor we assume a value of 0.975, for an annual basis. The technological parameter representing the output-business capital elasticity is assumed to be $\alpha = 0.35$. The preference parameter representing the weight of consumption in the utility function, $\gamma$, is fixed to 0.4. The distribution parameter of the CES for consumption and defense services is calibrated using data for military spending as a fraction of total consumption, fixing $\omega = 0.975$. Depreciation rates are taken from the Bureau of Economics Analysis (BEA). For the nonmilitary physical capital the depreciation rate is 7.24% per year, whereas for military physical capital it is slightly higher, at 9.02% per year. Finally, parameters values governing the stochastic processes for both aggregate productivity and ISTC are also standard in the literature, and for all stochastic processes we assume an autoregressive parameter of 0.90 and a standard deviation for the random component of 0.01. Table 1 summarizes the external calibration of the model.

**Table 1. External calibration.**

| Parameter | Definition | Value | Source |
|---|---|---|---|
| Preferences | | | |
| $\gamma$ | Consumption weight | 0.4000 | Standard |
| $\omega$ | Consumption distribution | 0.9750 | BEA |
| $\beta$ | Discount factor | 0.9875 | Standard |
| Technology | | | |
| $\alpha$ | Output-business capital elasticity | 0.3500 | BLS |
| $\phi$ | Adjustment cost | 2.000 | Standard |
| $\delta_k$ | Business capital depreciation rate | 0.0724 | BEA |
| $\delta_m$ | Military capital depreciation rate | 0.0902 | BEA |

## Results and discussion

### Steady-state and internal calibration

Data for capital stock and investment are taken from the BEA for the period 2002–2018. Average total investment is about 21.53% of GDP, split between a nonmilitary investment ratio of 0.2058 and a military investment ratio of 0.095. Steady state output has been normalized to 100 ($\bar{Y} = 100$). Given these figures, total nonmilitary capital stock is $\bar{K} = \left(\frac{0.2058}{0.0724}\right) \times 100 = 284.25$. Military capital stock is $\bar{X} = \left(\frac{0.0095}{0.0902}\right) \times 100 = 10.53$. These values mean that military capital in the US is around 10% of GDP, and about 3.57% of total capital stock.

According to the International Institute of Strategic Studies, the worldwide proportion of military labor over total labor (armed forces personnel over total labor force) was 0.804 in 2017, showing a very flat decreasing trend over the period 1990–2017. The figure for the US in 2017 was 0.88 (a force of 1,380,895), similar to the world average. If the fraction of working hours over total discretionary available time is around 0.33 (see BLS), and taking into account a total labor force of about 157 million, this implies that the fraction of hours used in the defense production sector is:

$$\bar{S} = 0.33 \times \left(\frac{1,380,895}{157,000,000}\right) = 0.0029$$

whereas the fraction of nonmilitary working hours is 0.314.

The internally calibrated parameters are estimated based on the equations of the model in steady state. We use this procedure to structurally estimate the three key parameters of the model: the elasticity of substitution between private consumption and defense services, the distributional parameter of the CES military production function, and the elasticity of substitution between military capital and military personnel. Therefore, we solve a system of 12 equations for 12 unknowns ($C, D, K, X, I_k, I_m, L, S, M, \eta, \mu, \phi$). Therefore, we add the following three additional equations to the system:

$$\bar{L} = 0.3271$$

$$\bar{S} = 0.0029$$

$$\bar{I}_m / {\bar{Y}} = 0.0095$$

**Table 2. Internal calibration and target moments.**

| Parameter | Definition | Value | Target moment |
|---|---|---|---|
| Preferences | | | |
| $\eta$ | Consumption/defense elasticity of substitution | -0.7992 | Non-military labor |
| Technology | | | |
| $\varphi$ | Military capital/military personnel | 0.4558 | Military capital to output |
| $\mu$ | Distribution parameter | 0.212 | Military labor |

The internally calibrated parameters are jointly determined, but for clarity of exposition we associate each parameter with the particular target moment most closely identified by that parameter. Table 2 presents the results of the internal calibration. The estimated value for the parameter $\eta$ is -0.7992, which implies that the elasticity of substitution between private consumption goods and services and national defense services is 0.5558. This estimated value is well below one, meaning that both types of goods are complementary, and consistent with [18, 19]. On the other hand, the estimated elasticity of substitution between military capital and military personnel is 1.9775. This value of the elasticity of substitution is well above unity, as it is usually assumed for the nonmilitary sector. This means that substitution between capital and labor is easier in the military sector than in the rest of the economy, and that increasing military capital and reducing military personnel would be a pattern more accentuated in the military sector than in the rest of the economy. A possible explanation of this high estimated elasticity of substitution is that in the military sector only one final good, defense, is produced, whereas in the business sector, a large variety of final goods are produced with different technologies and different capital and labor intensities. Finally, the estimated distribution parameter of the defense CES function is 0.2086, a value inferior to the standard 0.35 used in the nonmilitary production function. This is a consequence of higher capital deepening in the military sector compared to the nonmilitary sector. Indeed, the observed capital/labor ratio is around a value of 9 in the nonmilitary sector, whereas it is about four times higher (around a value of 36) in the military sector. Given these estimates and recent technological progress in automation, robotization, and artificial intelligence, we expect further substitution of military personnel by military equipment, increasing capital deepening in the military sector over the nonmilitary sector, as the first only produces a single good: national defense.

## Neutral technological shocks

The main objective of this study is to investigate the relationship between the business economy and the military sector. For that, we simulate the calibrated model to assess how both sectors respond to different technological shocks. First, we study the response of both sectors to an idiosyncratic neutral technological shock. This is a relevant exercise, as no clear relationship between the military sector and the rest of the economy has been established in the literature [2, 17]. When a neutral technological shock affects the nonmilitary production sector, output, private consumption of goods and services, investment in business capital, and civil employment increase, replicating the results obtained in the standard real business cycle analysis (Fig 1). But our model is rich enough to also assess how the productivity shock in the business sector affects the military sector. Following this shock, the number of military personnel also increases in response to the higher productivity in the nonmilitary sector, although in a very small quantity, indicating that leisure reduces. We also observe an increase in military investment, which leads to an increase in the military capital stock. This increase in the quantity of inputs devoted to the military sector leads to an increase in the production of defense services.

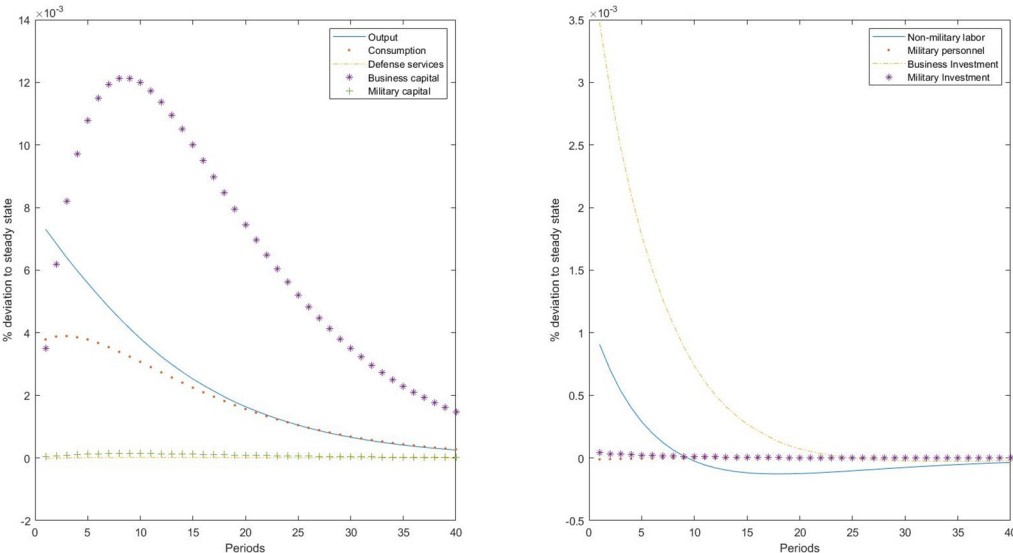

**Fig 1. Impulse-response to a business-neutral productivity shock.**

The underlying transmission mechanism for a productivity shock in the business sector to the military sector is explained by an income effect (a demand channel). The productivity shock in the business sector increases total output, and hence, more resources are available for consumption and investment. The higher level of resources is shared by the two sectors, increasing both private consumption and defense services, and increasing investment in non-military and military capital. Therefore, we find a positive impact of the economic activity on the military sector. The explanation behind this result comes from the estimated complementarity between private consumption and defense services. The higher the complementarity between the two goods, the higher the positive effect of the economy on military spending. However, this positive transmission mechanism from the business sector to the military sector does not operate in the case that private consumption and defense services are substitutes.

The conclusion is that the expansion of the economic activity results in a rise in the resources devoted to the production of defense only in the case that private consumption and defense services are complements. Indeed, our estimation indicates that they are indeed complements, resulting in the phenomenon that military spending is procyclical, as the higher the GDP, the higher the level of military expenditures. However, we also find that the proportion by which military spending increases is lower than the increase in final output, and hence, the military spending to output ratio is found to be countercyclical.

Second, we investigate the relationship in the other direction, that is, from the military sector to the rest of the economy. For that, we simulate an aggregate productivity shock to the defense production technology. This shock has several interpretations. For instance, it can represent a reduction in the external threat, thus increasing the efficiency of military expenditures in producing protection. This would also be interpreted as an improvement in doctrine and tactics, an increase in military personnel human capital, or the introduction of nuclear weapons that can provide national defense with fewer inputs. The main results of this simulation exercise are presented in Fig 2.

When the defense production technology is the one hit by a neutral productivity shock, we find a reduction in both military personnel and military investment. This is, in principle, a counterintuitive result, but it can be explained by the particular characteristic of the

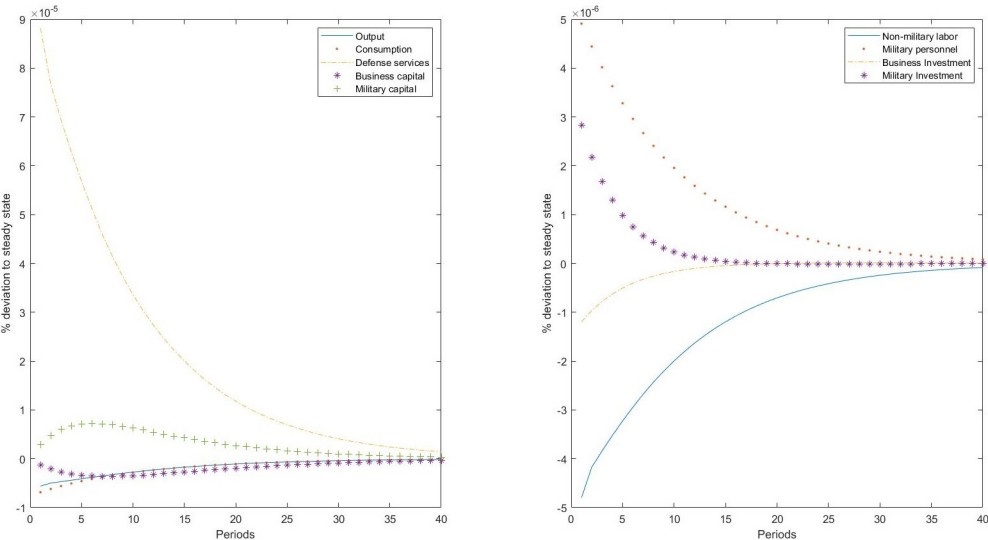

**Fig 2. Impulse-responses to a positive productivity shock to defense production.**

production of defense as a public good. The higher productivity in producing defense releases resources from the military sector that are reallocated in the rest of the economy. This reduction in military spending does not imply a reduction in the production of defense. On the contrary, production of defense services increases with the shock, indicating that the reduction in military inputs is a consequence of the optimal split of final output between the two sectors. Therefore, in this case we observe a substitution effect (a supply channel), reallocating productive factors from the military sector to the business sector. The military sector produces a single public good, defense services, and once the optimal level of defense is reached, no more resources are devoted to this sector. Importantly, we find that private consumption of goods and services also increases. This is a consequence of the increase in final output, as now the rest of the economy has more resources for business production. We observe a readjustment in investment and in labor, but in the opposite direction as one would expect for the case of a productivity shock. Military investment reduces and business investment increases in a portfolio readjustment. A similar adjustment is observed in labor, where the decline in military personnel is almost identical to the increase in nonmilitary labor. Therefore, the positive effect of the productivity shock to defense production is a direct consequence of the fact that more resources previously used for military spending are now not needed and reallocated to the business economy. This is consistent with previous results in the literature regarding the so-called "peace dividend" [22].

Notice that the response of the military sector to an aggregate productivity shock is completely different than the response we observe in the rest of the economy. A positive productivity shock to the military sector reduces military investment, and hence, military capital stock declines. However, the shock has a positive effect on the rest of the economy, increasing business investment, and hence, increasing business capital stock through a supply channel. A similar adjustment is observed in labor. Therefore, we find positive spillovers across the two sectors in both directions. However, transmission mechanisms differ. The positive effect of the military sector on the rest of the economy is not a consequence of a positive relationship between military spending and the rest of the economy; quite the contrary, it is a consequence of decreasing military spending. Defense services increase more than consumption, given that

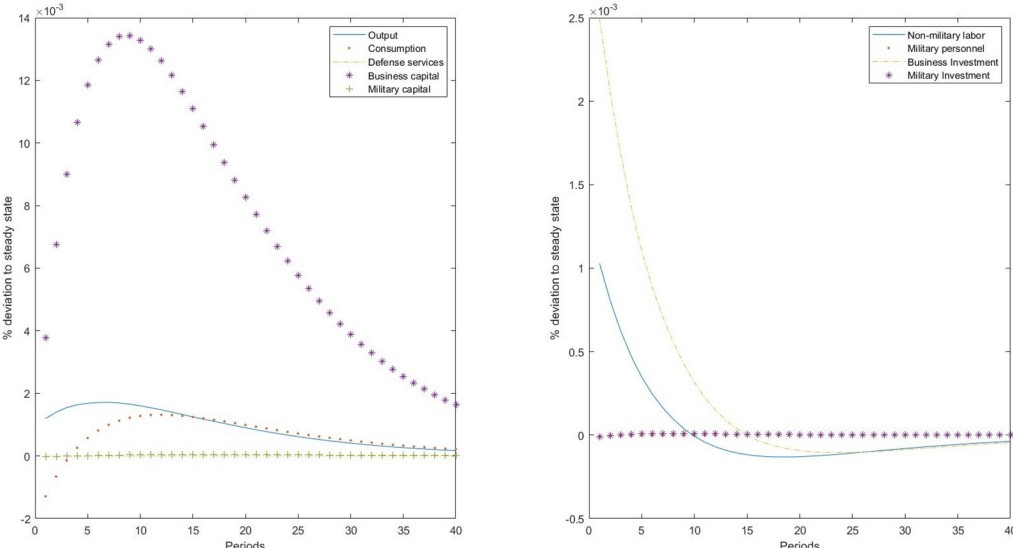

**Fig 3. Impulse-responses to an ISTC to the business sector.**

now the production of defense is more efficient, reducing the relative price of defense services with respect to private consumption of goods and services. Given the higher efficiency in the production of defense, fewer resources, both capital and military personnel, need to be devoted to the military sector. This explains why the shock has a positive effect on the rest of the economy. By contrast, the positive effect of the economy on the military sector is explained by an income effect.

## ISTC shock

ISTC to investment can be an important source for explaining output fluctuations and output growth [24]. Our model economy considers two types of physical capital: nonmilitary fixed assets and military fixed assets. The composition of these two types of capital, mainly for the case of equipment, is highly heterogeneous, and technological characteristics can be different. In general, we can assume that embodied technological progress in military equipment is different from that in nonmilitary equipment. This affects the relative prices of military versus nonmilitary capital assets over time. The question we want to answer here is the relationship between the military sector and the rest of the economy when the relative price of investment changes in each of the two sectors. For that, we study the effects of idiosyncratic ISTC shock to both the nonmilitary and military sectors.

First, Fig 3 plots the responses of the main variables of the model to an ISTC to business investment. The main consequence of this technological shock is a change in input prices. This shock reduces the relative price of business capital relative to both consumption (private consumption and defense services) and military capital. Productivity gains are only associated with the new capital invested, provoking an intertemporal substitution effect between business investment and consumption. Additionally, we observe an intratemporal substitution between consumption and leisure, leading to an increase in nonmilitary labor supply. Final output increases following the shock, showing a hump-shaped response. This is a consequence of the increase in investment which accumulates into business capital. As a consequence of the change in the relative price between consumption and investment, private consumption reduces initially.

Nevertheless, the ISTC shock to the business investment also has effects on the military sector, although very small in quantitative terms. The shock also changes the relative price of nonmilitary to military capital assets, but we do not observe a readjustment of investment by increasing business capital and reducing military capital, as should be the case if the economy is composed of different productive sectors. In impact, military investment decreases, although it recovers in the following periods. A similar behavior is observed in military personnel. Therefore, intertemporal and intratemporal substitution effects are observed in the rest of the economy, but not in the military sector. The most important result is that an idiosyncratic ISTC shock to business investment does not provoke a significant reduction in military investment, as would be expected, as military fixed assets are now more expensive relative to business fixed assets. Again, the explanation of this result has to be found in the particular characteristics of the military sector in producing a single public good which is of value for households. Indeed, the negative response of private consumption is not observed in defense services, which remain almost constant.

Next, we repeat the previous exercises, but considering an ISTC shock to the military sector. This could represent a policy directing a change in R&D of military equipment. ICTC to military investment would change the relative price of military capital assets to nonmilitary capital assets, but also with respect to private consumption. In principle, we could expect an increase in military investment and a decrease in business capital, with an overall negative effect on the economy. However, as shown in Fig 4, this is not the case. It is true that business investment decreases as a consequence of the shock, provoking a reduction in the business capital stock. However, we find the existence of a substitution effect between military inputs, resulting in a decline in working time devoted to defense production activities. The reason is that production of defense services has a reduction in impact (as does private consumption, given that consumption is now more expensive relative to investment), and given the higher level of military capital, the number of military personnel is reduced. After the shock, the response of defense services is positive, following a similar pattern to that of private consumption. Simultaneously, nonmilitary labor increases, resulting in a final positive effect on final output.

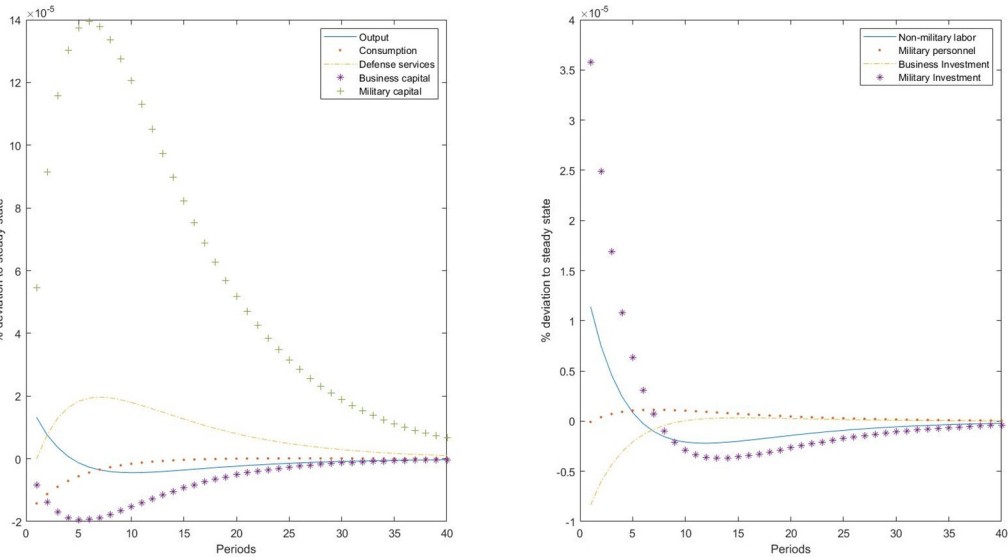

**Fig 4. Impulse-responses to an ISTC to the military sector.**

These findings are striking and consistent with the results of the internal calibration of the model. First, they are consistent with the data showing that the military capital to military personnel ratio is four times higher than the business capital to workers ratio. Technological progress reduces the relative price of military capital and incentives military investment. Given the optimal supply of defense services, which moves in the same direction as does private consumption given their complementarity, increases in military capital have to be compensated with declines in military personnel. Second, this can explain the previous estimations of the technological parameters of the military CES production function, where the elasticity of substitution between military capital and military personnel was found to be very high, and the distribution parameter representing the weight of military capital was found to be low relative to the value observed in the rest of the economy. Therefore, it is expected that ISTC to military equipment will accentuate the military capital to military personnel ratio in the defense production function. Finally, the ISTC shock to the military sector has an expansionary effect on the rest of the economy. This is a counterintuitive result, as the change in the relative price of investment between the two sectors would move investment from the business sector to the military sector, reducing final output. Although this substitution effect is observed, it is quantitatively small, and labor adjustment is enough to compensate the business capital decline and to increase final output.

## Conclusions

This study investigated the relationship between the military sector and the rest of the economy, using a two-sector DSGE model. We defined a technology for the military sector where both physical capital and labor are used in the production of national defense. The military sector is represented by the use of a fraction of the economy's resources devoted to produce national defense. National defense is a public good that produces utility for households. Internal calibration of the model reveals that the elasticity of substitution between private consumption of goods and services and defense services is very low, indicating that they are complementary. This complementarity between private consumption and defense services is key in assessing the relationship between the two sectors.

Studies in the literature have focused on investigating the relationship between military spending and economic growth. However, the results are mixed, and no consensus has emerged from a large number of empirical papers estimating the relationship between economic growth and military spending. Furthermore, recent evidence suggests no relationship between military expenditure and the rest of the economy [2]. This study contributes to this debate by investigating the relationship between both sectors in a general equilibrium environment where different shocks are considered. Our results indicate the existence of a positive bidirectional relationship between both sectors across the business cycle. The study identifies two different transmission mechanisms between the business and military sectors: an income effect (a demand channel) from the business sector to the military sector and a substitution effect (a supply channel) from the military sector to the business sector. First, defense services is a public good that produces utility for households, and hence, the government must devote some resources to the provision of this public good. The larger the total resources of the economy, the higher the military spending and the production of defense services, given the estimated complementarity between private consumption of goods and services and defense services. Second, any technological improvement in the military sector has positive effects on the rest of the economy, by releasing resources from the military sector to the business sector. This is true even in the case of an ISTC to investment in military equipment. Hence,

technological progress in the military sector releases resources that can be used in the rest of the economy.

In sum, the findings from this study indicate that the macroeconomy has a positive impact on the military sector and in the quantity of resources devoted to the production of national defense. Military spending is procyclical, but the fraction of GDP devoted to military spending is countercyclical. On the other hand, the military sector implies that a fraction of total resources has to be diverted to the production of defense services. Any positive shock affecting defense production technology will have a positive impact on the macroeconomy, although quantitatively it is of little value, given the small relative importance of the military sector on the whole economy. These positive spillovers are a consequence of the particular characteristics of the military sector in producing a single public good, which is a complement of private consumption.

## Acknowledgments

I would like to thank Ron P. Smith and the Editor for very helpful comments on a previous version of the manuscript.

## Author Contributions

**Conceptualization:** José L. Torres.

**Data curation:** José L. Torres.

**Formal analysis:** José L. Torres.

**Funding acquisition:** José L. Torres.

**Investigation:** José L. Torres.

**Methodology:** José L. Torres.

**Project administration:** José L. Torres.

**Resources:** José L. Torres.

**Software:** José L. Torres.

**Supervision:** José L. Torres.

**Validation:** José L. Torres.

**Visualization:** José L. Torres.

**Writing – original draft:** José L. Torres.

**Writing – review & editing:** José L. Torres.

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
