## [Decision Letter · Decision Letter 0]

7 Sep 2020

PONE-D-20-24181

The Production of Defence and the Macroeconomy

PLOS ONE

Dear Dr. Torres,

Thank you for submitting your manuscript to PLOS ONE. After careful consideration, we feel that it has merit but does not fully meet PLOS ONE’s publication criteria as it currently stands. Therefore, we invite you to submit a revised version of the manuscript that addresses the points raised during the review process.

We look forward to receiving your revised manuscript.

Kind regards,

Javier Ordóñez-Monfort

Academic Editor

PLOS ONE

Journal Requirements:

"The funders had no role in study design, data collection and analysis, decision to

publish, or preparation of the manuscript."

Reviewers' comments:

Reviewer's Responses to Questions

**Comments to the Author**

1. Is the manuscript technically sound, and do the data support the conclusions?

Reviewer #1: Yes

2. Has the statistical analysis been performed appropriately and rigorously? 

Reviewer #1: Yes

3. Have the authors made all data underlying the findings in their manuscript fully available?

Reviewer #1: Yes

4. Is the manuscript presented in an intelligible fashion and written in standard English?

Reviewer #1: Yes

5. Review Comments to the Author

Reviewer #1: The large empirical literature studying the relationship between military spending and output or growth shows no conclusive result. Partly this is because there is a difficult identification problem, military expenditure influences output through supply effects, but output influences military expenditure through demand effects, richer societies can afford to spend more on the military. Rather than trying to estimate the effect, this paper constructs a calibrated dynamic stochastic general equilibrium, DSGE, model, with capital and labour in military and civilian sectors.

The civilian sector is a stripped down (e.g. no inflation or interest rates) version of the sort of DSGE/real business cycle model widely used in Central Banks. Defence services are a public good, which appear in the utility function with consumption and leisure. Parameter values for the non-military sector are taken from the literature; for the military sector they are internally calibrated to match early 21st century US target values. It is a steady state model which means that there is no role for changing threats.

The results are dependent on the data and parameters. Military capital, and its rate of depreciation, is difficult to measure. So there must be considerable uncertainty about the estimated military capital-labour ratio which is four times the civilian capital-labour ratio. This is treated as data, but may be too high. A fighter aircraft and pilot is not typical of the capital labour balance of the whole armed forces.

The key military parameters are the elasticities of substitution between (a) consumption and defence services at 0.5558 and (b) military capital and personnel at 1.9775 and (c), the distributional parameter of the CES military production function. Elasticities of substitution are inherently difficult to measure, even for the non-military economy, which is why the Cobb-Douglas unit elasticity is so popular. While the consensus for the aggregate economy is that the elasticity is less than unity, there are some, like Piketty, who regard it as greater than unity.

One might question the high elasticity of substitution between military equipment and personnel. This would imply large changes of the share of the budget on personnel after a move from conscript to volunteer forces. This is not what has been observed. Typically the rise in wages is matched by a roughly equivalent reduction in numbers. There is also an issue of the normalization of a CES production function to removes the problem that labour and capital are measured in different units.

The paper may be better considered as an interesting piece of numerical theory, conditional on the parameters.

The paper considers how the variables respond over time to neutral and investment specific technology shocks and has some interesting results on the transmission mechanisms between the sectors. As the paper notes, not all the results are what one might expect. A positive neutral technology shock producing higher output gives more military spending and consumption. A positive technology shock to military investment, perhaps the introduction of nuclear weapons, can reduce military expenditure, since the desired defence can be provided at lower cost.

Although it is comprehensible, the paper would benefit from careful proofreading.

6. PLOS authors have the option to publish the peer review history of their article (what does this mean?). If published, this will include your full peer review and any attached files.

Reviewer #1: **Yes: **Ron P Smith

---

## [Author Response · Author response to Decision Letter 0]

17 Sep 2020

The Production of National Defence and the Macroeconomy

PONE-D-20-24181

Reviewer #1: The large empirical literature studying the relationship between military spending and output or growth shows no conclusive result. Partly this is because there is a difficult identification problem, military expenditure influences output through supply effects, but output influences military expenditure through demand effects, richer societies can afford to spend more on the military. Rather than trying to estimate the effect, this paper constructs a calibrated dynamic stochastic general equilibrium, DSGE, model, with capital and labour in military and civilian sectors.

Response: Thank you very much for your comments and your careful reading of my manuscript. I totally agree with you. Heterogeneous and non-conclusive results in the large empirical literature indicate the existence of an identification problem. Indeed, this is the starting point of my paper, that is, contributing to the literature by studying the relationship between military spending and output using a general equilibrium theoretical framework, where both demand and supply effects are considered. The existence of two different transmission channels is exactly what is observed in the model, where there is an income effect (a demand channel) from the non-military sector to the military sector, and a substitution effect (a supply channel) from the military sector to the rest of the economy. In my opinion the empirical literature should focus on using Vector Autoregressive (VAR) techniques, which can deal with the identification problem that seems to exist in previous empirical literature. 

The civilian sector is a stripped down (e.g. no inflation or interest rates) version of the sort of DSGE/real business cycle model widely used in Central Banks. Defense services are a public good, which appear in the utility function with consumption and leisure. Parameter values for the non-military sector are taken from the literature; for the military sector they are internally calibrated to match early 21st century US target values. It is a steady state model which means that there is no role for changing threats.

Response: I acknowledge that the model is a simplification of the world, but still useful to investigate the links between the two sectors. Threat is a key variable for the determination of military spending but in my model is assumed to be fixed, as the focus is to investigate the links between the production of defense services and the rest of the economy, and not optimal military spending given a threat.

The results are dependent on the data and parameters. Military capital, and its rate of depreciation, is difficult to measure. So there must be considerable uncertainty about the estimated military capital-labour ratio which is four times the civilian capital-labour ratio. This is treated as data, but may be too high. A fighter aircraft and pilot is not typical of the capital labour balance of the whole armed forces.

Response: You are right. Results are conditioned to the calibration of the parameters of the model. For the nonmilitary sector I use standard values used in the literature. However, for the military sector, given that little information is available, I use an internal calibration approach to estimate the key parameter of the model conditional to data on military spending, military capital and military personnel. I agree with you that physical capital is difficult to measure. Indeed, this is one of the main reasons why capital stock is not measured in National Accounts, and only capital formation and capital consumption are considered. Nevertheless, capital and depreciation rates are difficult to measure not only in the military sector, but also in the non-military sector. However, for the US we have high quality data on military capital and depreciation rates. Military capital and depreciation rate are taken from the BEA (Bureau of Economic Analysis). In BEA data, military capital and depreciation rates are estimated for the following assets: Equipment, Aircraft, Missiles, Ships, Vehicles, Electronics, Other equipment, Structures, Buildings, Residential, Industrial, Military facilities, Intellectual property products, Software, and Research and development. Given the characteristics of military capital, a capital-labor ratio in the military sector four times that of the civilian sector seems plausible. Finally, although not shown in the paper, I have carried out a sensitivity analysis by changing the values of the parameters to check the robustness of the model, but results remain without significant changes, except for the case of the elasticity of substitution between final goods and defense services, where the income effect declines but the substitution effect increases as we reduce the complementarity between both types of goods.

The key military parameters are the elasticities of substitution between (a) consumption and defence services at 0.5558 and (b) military capital and personnel at 1.9775 and (c), the distributional parameter of the CES military production function. Elasticities of substitution are inherently difficult to measure, even for the non-military economy, which is why the Cobb-Douglas unit elasticity is so popular. While the consensus for the aggregate economy is that the elasticity is less than unity, there are some, like Piketty, who regard it as greater than unity.

Response: This is related to the previous comment. Given that no information is available about technological parameter of the defense production function, the strategy in my paper was to estimate that parameters using an internal calibration approach. Standard macroeconomic models assume a unitary elasticity of substitution between capital and labor for the aggregate economy. The main reference here is Chirinko (Chirinko, R.S. (2008). Sigma: The long and short of it. Journal of Macroeconomics, 30, 671-686.). In the literature, some authors argue that the elasticity of substitution between labor and capital is lower than one, i.e., labor and physical capital are complement, whereas a few works supports an elasticity of substitution above one. 

One might question the high elasticity of substitution between military equipment and personnel. This would imply large changes of the share of the budget on personnel after a move from conscript to volunteer forces. This is not what has been observed. Typically the rise in wages is matched by a roughly equivalent reduction in numbers. There is also an issue of the normalization of a CES production function to removes the problem that labour and capital are measured in different units.

Response: The estimated elasticity of substitution between military capital and personnel could be considered high compared to the values usually used for the aggregate economy, but is a value estimated using data for the military sector. However, Chirinko (2008) reports values from 0 to 3.4. On the other hand, several authors report that the elasticity of substitution for particular sectors can be large than one. For instance, Tevlin and Whelan (Tevlin, S. and Whelan, K. (2003). Explaining the investment boom of the 1990s. Journal of Money, Credit and Banking, 35, 1-22) find an aggregate elasticity of 0.18, but a much larger value of 1.59 for computers. Similarly, for the UK, Bakhshi et al. (Bakhshi, H., Oulton, N. and Thompson, J. (2003). Modelling investment when relative prices are trending: Theory and evidence for the United Kingdom. Bank of England Working Paper No. 189.), report estimates of 0.32 for the whole economy and 1.33 for computers. Therefore, an estimated elasticity of substitution of 1.98 for the military sector is plausible. On the other hand, the approach used in the paper overcomes the problem associated to the normalization of a CES function as the distribution parameter of the CES and the elasticity of substitution parameter are estimated simultaneously, which it is equivalent to the normalization of a CES function, as indicated by de La Grandville (de La Grandville, O. (2016). Economic Growth-A Unified Approach. Cambridge University Press.)

The paper may be better considered as an interesting piece of numerical theory, conditional on the parameters.

Response: You are right. Results are conditioned to the calibration of the parameters of the model, as I indicated above. This is the main reason to use an internal calibration approach for estimating technological parameters of the defense production function, using data on military spending, military capital and military personnel. This estimation approach produces structural estimated values for the key parameters of the model, based on data.

The paper considers how the variables respond over time to neutral and investment specific technology shocks and has some interesting results on the transmission mechanisms between the sectors. As the paper notes, not all the results are what one might expect. A positive neutral technology shock producing higher output gives more military spending and consumption. A positive technology shock to military investment, perhaps the introduction of nuclear weapons, can reduce military expenditure, since the desired defence can be provided at lower cost.

Response: I would like to add that the main advantage of the approach used in the paper is that we can identify “general equilibrium” effects, including both demand and supply effects, between both sectors. I think this could contribute to a better understanding of the relationship between the military sector and the rest of the economy.

Although it is comprehensible, the paper would benefit from careful proofreading.

Response: The revised manuscript has been professionally proofread in order to avoid grammatical errors and to improve clarity and ease of reading. Many thanks again for your careful reading of my manuscript.

---

## [Decision Letter · Decision Letter 1]

24 Sep 2020

The production of defense and the macroeconomy

PONE-D-20-24181R1

Dear Dr. Torres,

We’re pleased to inform you that your manuscript has been judged scientifically suitable for publication and will be formally accepted for publication once it meets all outstanding technical requirements.

Kind regards,

Javier Ordonez, Ph.D.

Academic Editor

PLOS ONE

Additional Editor Comments (optional):

Reviewers' comments:

Reviewer's Responses to Questions

**Comments to the Author**

1. If the authors have adequately addressed your comments raised in a previous round of review and you feel that this manuscript is now acceptable for publication, you may indicate that here to bypass the “Comments to the Author” section, enter your conflict of interest statement in the “Confidential to Editor” section, and submit your "Accept" recommendation.

Reviewer #1: All comments have been addressed

2. Is the manuscript technically sound, and do the data support the conclusions?

Reviewer #1: (No Response)

3. Has the statistical analysis been performed appropriately and rigorously? 

Reviewer #1: (No Response)

4. Have the authors made all data underlying the findings in their manuscript fully available?

Reviewer #1: (No Response)

5. Is the manuscript presented in an intelligible fashion and written in standard English?

Reviewer #1: (No Response)

6. Review Comments to the Author

Reviewer #1: (No Response)

7. PLOS authors have the option to publish the peer review history of their article (what does this mean?). If published, this will include your full peer review and any attached files.

Reviewer #1: **Yes: **Ron Smith

---

## [Editor Report · Acceptance letter]

1 Oct 2020

PONE-D-20-24181R1 

The production of national defense and the macroeconomy 

Dear Dr. Torres:

I'm pleased to inform you that your manuscript has been deemed suitable for publication in PLOS ONE. Congratulations! Your manuscript is now with our production department. 

Kind regards, 

on behalf of

Dr. Javier Ordonez 

Academic Editor

PLOS ONE